# Optimal Efficiency-Envy Trade-Off via Optimal Transport

**Steven Yin**
Department of Industrial Engineering and Operations Research
Columbia University
New York, NY 10027
sy2737@columbia.edu

**Christian Kroer**
Department of Industrial Engineering and Operations Research
Columbia University
New York, NY 10027
christian.kroer@columbia.edu

## Abstract

We consider the problem of allocating a distribution of items to $n$ recipients where each recipient has to be allocated a fixed, prespecified fraction of all items, while ensuring that each recipient does not experience too much envy. We show that this problem can be formulated as a variant of the semi-discrete optimal transport (OT) problem, whose solution structure in this case has a concise representation and a simple geometric interpretation. Unlike existing literature that treats envy-freeness as a hard constraint, our formulation allows us to *optimally* trade off efficiency and envy continuously. Additionally, we study the statistical properties of the space of our OT based allocation policies by showing a polynomial bound on the number of samples needed to approximate the optimal solution from samples. Our approach is suitable for large-scale fair allocation problems such as the blood donation matching problem, and we show numerically that it performs well on a prior realistic data simulator.

## 1   Introduction

In this work, we focus on the problem of finding an allocation policy that divides a pool of items, represented by a distribution $\mathcal{D}$, to $n$ recipients, under the constraint that each recipient $i$ must be allocated a pre-specified fraction $p_i^*$ of the items, where $p_i^* \in (0,1), 1^\top p^* = 1$ is an input to the problem that characterizes the priority of each recipient. We refer to $\{p_i^*\}_{i=1}^n$ as the target matching distribution. In addition to this matching distribution constraint, we also require that the recipient's envy, which we will define formally later, be bounded. Unlike the existing resource allocation literature, where envy-free is either treated as a hard constraint or not considered at all (in divisible settings), we allow the central planner to specify the level of envy that is tolerated, and find the most efficient allocation given the amount of envy budget. This allows us to reduce the envy significantly without paying the full price of fairness with respect to efficiency.

To concretely motivate our model, let us consider the blood donor matching problem that was first studied in [21]. The Meta platform has a tool called *Facebook Blood Donations*, where users who opt in to receive notifications are notified about blood donation opportunities near them. Depending on the user's and the blood bank's specific characteristics (e.g., age, occupation for the user, and locations, hours for the blood bank), notifications about different donation opportunities have different

36th Conference on Neural Information Processing Systems (NeurIPS 2022).

probabilities of resulting in an actual blood donation. The platform would like to send each user the most relevant notifications (to maximize the total number of potential blood donations), while maintaining certain fairness criteria for all the blood banks that participate in this program. Although the platform can theoretically send each user multiple notifications about multiple blood banks, for user experience and other practical reasons, this is not done, at least in the model introduced in [21]. Therefore in this problem, users' attention is the scarce resource that the platforms needs to allocate to different blood banks. The most natural type of fairness criteria in this setting is perhaps the number of users that received notifications about each of the blood banks. For example, it is not desirable to match zero users to a particular, potentially inconveniently-located, blood bank, even if matching zero users to this blood bank results in more blood donations in expectation.

Another fairness desideratum commonly studied in the literature is called *envy-freeness*. We say that recipient $A$ envies recipient $B$ if $A$ values $B$'s allocation more than her own. Intuitively, an allocation that is envy-free–where no agent envies another agent–is perceived to be fair.

This paper considers both of the two fairness criteria mentioned above: we study a setting where the goal is to maximize social welfare under a matching distribution constraint, while ensuring that each recipient has bounded envy. We make the following contributions in regards to this problem:

1. We formulate it as a constrained version of a semi-discrete optimal transport problem and show that the optimal allocation policy has a concise representation and a simple geometric structure. This is particularly attractive for large-scale allocation problems, due to the fast computation of a match given an item. This insight also shines new light on the question of when envy arises, and when the welfare price on envy-freeness is large.

2. We propose an efficient stochastic optimization algorithm for this problem and show that it has a provable convergence rate of $O(1/\sqrt{T})$.

3. We investigate the statistical properties of the space of our optimal transport based allocation policies by showing a Probably Approximately Correct (PAC)-like sample complexity bound for approximating the optimal solution given finite samples.

In Section 3 we formally define the problem we are interested in. In Section 4, we show that this problem can be formulated as a semi-discrete optimal transport problem, whose solution has a simple structure with a nice geometric interpretation. Section 5 develops a practical stochastic optimization algorithm. In Section 6, we show that an $\epsilon$-approximate solution can be found with high probability given $\tilde{O}(\frac{n}{\epsilon^2})$ samples, where $n$ is the number of recipients. Finally, in Section 7 we demonstrate the effectiveness of our approach using both artificial and a semi-real data.

## 2    Literature Review

**Blood donation matching.**    McElfresh et al. [21] introduced this problem and modeled it as an online matching problem, where the matching quality between an user and a blood bank is assumed to be known to the platform. The model formulation there is complex and their matching policy is rather cumbersome, requiring a separate parameter for each (donor, recipient) pair. Compared to their paper, we are able to provide better structural insights to the problem by utilizing a simpler model that still captures the most salient part of the problem.

**Online Resource Allocation.**    Another strand of work that our paper is closely related to is that of online resource allocation, especially those with i.i.d. or random permutation input models. Agrawal, Wang, and Ye [2] studied the setting with linear objective and gave competitive ratio bounds. Then, Agrawal and Devanur [1] generalized the results to concave objectives and convex constraints. Later, Devanur et al. [14] improved the approximation ratio bounds and relaxed the input assumptions on the budgets. Balseiro, Lu, and Mirrokni [6] show that online mirror descent on the dual multipliers does well under both i.i.d. adversarial, and certain non-stationary input settings. However, none of theses papers study the envy-free criterion. Recently, Balseiro, Lu, and Mirrokni [5] studied an online resource allocation problem with fairness regularization. Although the authors did not explicitly study envy regularization, their regularization framework can be modified to accommodate envy regularization. However, like all the other papers mentioned in this paragraph, the offline solution is used as the benchmark to measure regret, but no explicit solution is given to the offline problem. Our analysis focuses on the offline problem, and draws an explicit connection to optimal transport,

which allowed us to provide a novel PAC-like analysis on the sample complexity of the problem. In another recent paper, Sinclair, Banerjee, and Yu [27] studied the trade-off between minimizing envy and minimizing waste, which refers to un-allocated resources. Despite close similarity between our titles, their offline benchmark is the standard Eisenberg-Gale program, which is envy-free, but does not address the welfare cost of achieving envy-freeness.

**Fair Division.** Envy is a popular concept studied in the fair division literature. A large body of these papers are formulated as a cake-cutting problem ([24, 12, 23]) where the resources are modeled as an interval and the agents' valuations are represented as functions on this interval. Caragiannis et al. [10] provide a analysis on the worst case efficiency loss due to the envy-freeness constraint. Later Cohler et al. [13] design algorithms for computing optimal envy-free cake cutting allocations under different relatively simple classes of valuation functions. Unlike these papers that focus on the hard constraint of zero envy, we treat the allowable envy as a parameter, and find the most efficient solutions subject to the desired amount of envy. In indivisible settings, there are some related concepts of envy that does not require a strict zero envy. For instance, Budish [9] proposes an approximate competitive equilibrium from equal incomes approach that achieves envy free up to 1 item (EF1). Caragiannis et al. [11] introduce the concept of envy free up to the Least Valued Good, which is a stronger version of EF1. Although these papers do allow a small amount of envy, these concepts are introduced mainly to circumvent the impossibilities introduced by the indivisible settings, not to allow central planners to control the level of envy tolerance. Finally, Donahue and Kleinberg [15] also studies a setting where the central planner can set his own tolerance of (un)fairness. However their setting is quite different from ours, as they consider the allocation of fixed amount of identical resource, where as we assume that recipients have different valuations for the different items.

**Optimal Transport.** OT has been applied before in resource allocation settings in the economics literature (see [17] for a survey). For example, the Hotelling location model with a continuous mass of consumers can be solved with the same assignment procedure as the one we consider for our allocation problem *without* envy constraints. The more general method of assigning points in space to a finite set of sites via this procedure was developed by Aurenhammer, Hoffmann, and Aronov [4]. Scetbon et al. [25] consider an OT setting with equitability (every agent's final utility is the same), which is a different fairness criteria from envy, and also not adjustable like in our setting. To the best of our knowledge, no existing application of OT models the envy constraints that we consider here.

## 2.1 Background on Optimal Transport

Since our work draws an explicit connection to optimal transport (OT), we provide a summary of key OT results here. Let $\alpha, \beta$ be two probability measures on the metric spaces $\mathcal{X}, \mathcal{Y}$ respectively. We define $\Pi(\alpha, \beta)$ as the set of joint probability measures on $\mathcal{X} \times \mathcal{Y}$ with marginals $\alpha$ and $\beta$. The *Kantorovich formulation* of the optimal transport problem [19] can be written as

$$L(\alpha, \beta) := \min_{\pi \in \Pi(\alpha, \beta)} \int_{\mathcal{X} \times \mathcal{Y}} c(x, y) d\pi(x, y) \tag{1}$$

where $c(x, y)$ is the cost associated with "moving" $x$ to $y$. This is called a transportation problem because the conditional probability $\pi(y|x)$ specifies a transportation plan for moving probability mass from $\mathcal{X}$ to $\mathcal{Y}$. Note that $\pi(x, y) = \pi(y|x) d\alpha(x)$. If $\beta$ is a discrete measure, i.e. $\mathcal{Y}$ is finite, then it is known [3] that the dual to (1) can be written as (here we abuse the notation $\beta$ to also represent the vector of probability masses, where $\beta_i$ is the probability mass on point $y_i$):

$$\max_{g \in \mathbb{R}^n} \mathcal{E}(g) := \sum_{i \in [n]} \left[ \int_{\mathbb{L}_{y_i}(g)} c(x, y_i) - g_i \, d\alpha(x) \right] + g^\top \beta \tag{2}$$

where $n = |\mathcal{Y}|$, and $\mathbb{L}_{y_i}$ is what is sometimes referred to as the *Laguerre cell*:

$$\mathbb{L}_{y_i}(g) = \{x \in \mathcal{X} : \forall i \neq j, c(x, y_i) - g_i \leq c(x, y_j) - g_j\} \tag{3}$$

**Proposition 1** (Proposition 2.1 [3])**.** *If $\alpha$ is a continuous measure, and $\beta$ a discrete measure, then $L(\alpha, \beta) = \max_g \mathcal{E}(g)$, and the optimal solution $\pi$ of (1) is given by the partition $\{\mathbb{L}_{y_i}(g^*), i \in [n]\}$, i.e. $d\pi(x, y_i) = d\alpha(x)$ if $x \in \mathbb{L}_{y_i}(g^*)$, 0 otherwise.*

# 3 Problem Formulation

There is a set of $n$ recipients $\mathcal{Y}$. There is a "pool" of items, represented by a distribution $\alpha$ over $\mathcal{X} \subseteq [0, \bar{x}]^n$. Each random draw from this distribution $X \sim \alpha$ is a vector representing the $n$ recipients' valuations of this item. The goal is to maximize the expected matched utilities of the recipients, while maintaining the constraint that the recipient $y_i$ is matched $p_i^*$ fraction of the times in expectation. Here $\{p_i^*\}_{i=1}^n$ is called the target matching distribution, which intuitively represents recipients' importance. Note that the constrains here are satisfied *in expectation*, which are sometimes referred to as "ex-ante" guarantees. The reason why we consider ex-ante guarantees has to do with the type of application we're interested in. Our motivating example is concerned with recommending hundreds of millions of users to different blood banks. In such settings, even if we want the constraints to hold ex-post, the large-scale nature of the problem and the law of large numbers means that in-expectation guarantees translate into something that is very close to holding ex-post. That is why for internet platform problems, requiring constraints to hold in expectation is a standard setup (see for example the literature on budget constraints in ad auctions, where this is the case [7]). A matching policy $\pi$ takes a valuation vector and maps it (potentially with randomness) to one of the $n$ recipients. Let $\pi(y|x)$ denote the probability of matching the item to $y$ given valuation vector $x$. The basic problem formulation is to solve the following optimization problem:

$$\max_\pi \mathbb{E}_{X \sim \alpha} \left[ \sum_{i=1}^n X_i \pi(y_i|X)] \right] \tag{4}$$
$$s.t. \quad \mathbb{P}\left[\pi(y_i|X)\right] = p_i^* \quad \forall i \in [n]$$

WLOG, we assume that $p_i^* > 0$ for all $i$. An example of such problem can be see in Figure 1, where $\alpha$ is a distribution over the unit square, and the goal is to partition the square into blue and orange regions (given to $A$ and $B$ respectively) such that each region covers the desired $p_A^*, p_B^*$ probability mass. Note that the orange and blue regions are allowed to over lap (probabilistic partition), and that the boundary does not have to be linear as illustrated in the figure. As we will show later in Section 4, despite the large design space permitted by the formulation in (4), we can in fact focus on a much smaller design space.

In resource allocation problems, it is often the case that we care not just about efficiency (the sum of all recipients' utilities), but also other fairness criteria. One of the most commonly studied fairness criteria is envy-freeness. Agent $y_i$ envies another agent $y_j$ if agent $y_i$ values the allocation given to $y_j$ more (after adjusting for their priority weights). We can formally define agent $y_i$'s envy as

$$Envy(y_i) = \max_j \mathbb{E}_\alpha \left[ X_i \pi(y_j|X) \frac{p_i^*}{p_j^*} - X_i \pi(y_i|X) \right] \tag{5}$$

Instead of the vanilla formulation in (4), we consider the following more general formulation:

$$\max_\pi \mathbb{E}_{X \sim \alpha} \left[ \sum_{i=1}^n X_i \pi(y_i|X)] \right] \tag{6}$$
$$s.t. \quad \mathbb{P}\left[\pi(y_i|X)\right] = p_i^* \quad \forall i \in [n]$$
$$Envy(y_i) \leq \lambda_i \quad \forall i$$

Most existing literature focuses on finding allocations such that $Envy(y_i)$ is at most 0 for every $y_i$. This can be a very restrictive constraint, often satisfied at the cost of reducing efficiency by a significant amount (This reduction is sometimes referred to as the Cost-of-Fairness). We take a different approach, and allow the central planner to set non-negative constraints on envy. Note that since we are motivated by internet-scale problems such as allocating hundreds of millions of users to donation centers, we focus on the ex-ante guarantees on the constraints.

# 4 Optimal Solution Structure

The space of feasible solutions for (6) is large, which makes the problem difficult to optimize directly. However we can use the tools from OT to reduce the search space to something with much more structure. The key observation is that (6) can be formulated as variation of the semi-discrete optimal transport problem given in Equation (1).

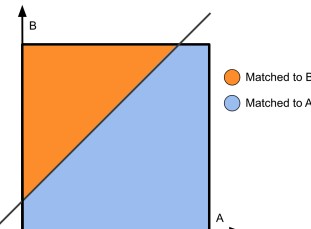 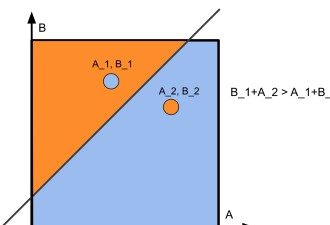

Figure 1: Left: An illustration of what Laguerre cells look like when $n = 2$. Consider any distribution on the support $[0, 1]^2$. The optimal division of the space is to move the diagonal line up or down until the probability mass contained in the orange region is equal to $p^*$. Right: A pictorial proof of the optimality of such partition. Suppose one can find an $\epsilon$ mass above this line that is matched to $A$, and an $\epsilon$ mass below the line that is matched to $B$, then switching the assignments of these two regions increases the matched weights because $B_1 + A_2 > A_1 + B_2$.

Let's first consider the simpler case in (4) where there are no envy constraints. In this case, the problem can be stated in the form of (1) as follows: the cost function is the negative utility of the matched recipient $c(x, y_i) = -x_i$, the $\beta$ measure is the discrete measure $\sum_{i=1}^{n} p_i^* \delta_{y_i}$, and the matching policy $\pi(y|x)$ in (6) is exactly the conditional probability of the joint distribution in (1). From Theorem 1 it follows that the optimal matching policy is represented by Laguerre cells given in (3): $x$ is matched to $y_i$ if $i = \arg\min_k -x_k - g_k$. Note that the dual variables $g \in \mathbb{R}^n$ serve as an "adjustment" over the agents' reported utilities, and the resulting matching policy is simply a greedy policy over this adjusted valuation vector.

Geometrically, each Laguerre cell is simply the intersection of half-spaces: $\mathbb{L}_i(g) = \cap_k \{x : x_i + g_i \geq x_k + g_k\}$. To visualize this better, consider the simple setting with two recipients $A, B$ where their valuations for an item is a joint distribution supported on $[0, 1]^2$. Suppose we want to match $p^*$ fraction of the items to recipient $B$. Figure 1 gives a proof-by-picture that the optimal strategy is to divide the space up with a slope-1 diagonal line such that the probability mass lying above the line is equal to $p^*$. This geometric interpretation of the matching policy plays a crucial role in getting us a sample complexity bound later in Section 6.

With this geometric interpretation of the solution space in mind, let us consider the more general case with envy constraints as formulated in (6). The envy constraints can be added to the OT problem in (1) like so:

$$L(\alpha, \beta, \lambda) = \min_{\pi \in \Pi(\alpha, \beta)} \int_{\mathcal{X} \times \mathcal{Y}} c(x, y) d\pi(x, y) \tag{7}$$

$$s.t. \int_{\mathcal{X}} c(x, y_j) d\pi(x, y_j) - \int_{\mathcal{X}} c(x, y_j) d\pi(x, y_k) \frac{\beta_j}{\beta_k} \leq \lambda_j \quad \forall (j, k) \in [n]^2, j \neq k$$

Although envy constraints make the solution space more complicated, we show that it retains the geometric structure of being the intersection of half-spaces. The dual of (7) can be derived using Fenchel-Rockafellar's theorem:

$$\max_{g \in \mathbb{R}^n, \gamma \in \mathbb{R}_+^{n^2 - n}} \mathcal{E}(g, \gamma) := \sum_{j \in [n]} \int_{\mathbb{L}_{y_j}(g, \gamma)} \bar{g}_{\gamma, c}(x, y_j) d\alpha(x) + g^\top \beta - \sum_{j, k, j \neq k} \gamma_{jk} \lambda_j \tag{8}$$

where

$$\bar{g}_{\gamma, c}(x, y_j) := \left(1 + \sum_{k \neq j} \gamma_{jk}\right) c(x, y_j) - \sum_{k \neq j} \gamma_{kj} c(x, y_k) \frac{\beta_k}{\beta_j} - g_j, \tag{9}$$

$$\mathbb{L}_y(g, \gamma) := \left\{x \in \mathcal{X} : y = \arg\min_{y' \in \mathcal{Y}} \bar{g}_{\gamma, c}(x, y')\right\}. \tag{10}$$

**Theorem 1.** *If $\alpha$ is a continuous measure, and $\beta$ a discrete measure, then $L(\alpha, \beta, \lambda) = \max_{g, \gamma} \mathcal{E}(g, \gamma)$, and the optimal solution $\pi$ of (7) is given by the partition $\{\mathbb{L}_{y_i}(g^*, \gamma^*), i \in [n]\}$: $d\pi(x, y_i) = d\alpha(x)$ if $x \in \mathbb{L}_{y_i}(g^*, \gamma^*)$, 0 otherwise.*

Note that when $c(x, y_i) = -x_i$, $\bar{g}_{\gamma,c}(x, y_j)$ is linear in $x$, which means that the new Laguerre cells $\mathbb{L}_y(g, y)$ given in Equation (10) are still intersections of half spaces (some examples are given later in Figure 2). Furthermore, the allocation policy can be interpreted as a greedy policy based on the adjusted utility given by (9), which contains additional interaction terms that take envy into account.

## 5    Stochastic Optimization

In Section 4 we showed that the optimal allocation policy to our problem has a simple geometric structure in the form of Laguerre cells. In this section we present a practical algorithm for actually computing the optimal Laguerre cells. First we show that the objective function $\mathcal{E}(g, \gamma)$ in (6) is concave by rewriting the objective as follows:

$$\mathcal{E}(g, \gamma) = \int_{\mathcal{X}} \min_{i \in [n]} \bar{g}_{\gamma,c}(x, y_i) d\alpha(x) + g^{\top}\beta - \sum_{j,k,j \neq k} \gamma_{jk}\lambda_j \tag{11}$$

Since $\bar{g}_{\gamma,c}(x, y_j)$ is linear in $g$ and $\gamma$ and taking a minimum preserves concavity, the objective function is concave. Therefore, the dual problem is a constrained convex optimization problem. The gradient of $\mathcal{E}(g, \gamma)$ can be computed as follows:

$$\nabla_g \mathcal{E}(g, \gamma)_j = -\int_{\mathbb{L}_{y_j}(g,\gamma)} d\alpha(x) + \beta_j \tag{12}$$

$$\nabla_\gamma \mathcal{E}(g, \gamma)_{jk} = \int_{\mathbb{L}_{y_j}(g,\gamma)} c(x, y_j) d\alpha(x) - \int_{\mathbb{L}_{y_k}(g,\gamma)} c(x, y_j)\frac{\beta_j}{\beta_k} d\alpha(x) - \lambda_j \tag{13}$$

---

**Algorithm 1:** Projected SGD for Envy Constrained Optimal Transport

**Input:** Distribution $\alpha$, target matching distribution $p^*$, timesteps $T$.

1  Initialize $g_0 = 0, \gamma_0 = 0, \eta = \frac{1}{\sqrt{T}}$.

2  **for** $t \leftarrow 0, 1, 2, \ldots, T$ **do**

3       Sample $x_t \sim \alpha$

4       $g_{t+1} \leftarrow g_t + \eta\hat{\nabla}_g\mathcal{E}(g, \gamma)$

5       $\gamma_{t+1} \leftarrow \left(\gamma_t + \eta\hat{\nabla}_\gamma\mathcal{E}(g, \gamma)\right)^+$

6  **return** $\sum_{t=1}^{T} g_t/T, \sum_{t=1}^{T} \gamma_t/T$

---

Calculating this gradient is hard, as it involves integration over an arbitrary measure $\alpha$. However, an unbiased, stochastic version of the gradient can be easily obtained from a single sample $x \sim \alpha$:

$$\hat{\nabla}_g\mathcal{E}(g, \gamma)_j = -\mathbb{1}[x \in \mathbb{L}_{y_j}(g, \gamma)] + \beta_j \tag{14}$$

$$\hat{\nabla}_\gamma\mathcal{E}(g, \gamma)_{jk} = c(x, y_j)\mathbb{1}[x \in \mathbb{L}_{y_j}(g, \gamma)] - c(x, y_j)\frac{\beta_j}{\beta_k}\mathbb{1}[x \in \mathbb{L}_{y_k}(g, \gamma)] - \lambda_j \tag{15}$$

The details of the algorithm is given in Algorithm 1. Standard projected SGD analysis (see for example [18]) tells us that Algorithm 1 converges at the rate $\mathcal{E}(g^*, \gamma^*) - \mathcal{E}(\mathbb{E}[g_T], \mathbb{E}[\gamma_T]) \leq O\left(\frac{1}{\sqrt{T}}\right)$.

## 6    Learning from Samples

So far we have assumed that the true underlying distribution is known, and that we can freely draw independent samples it. In many settings, we only have access to $\alpha$ in the form of finite number of i.i.d. samples. The goal of this section is to establish a sample complexity bound for solving the dual problem (8).

In this section, we focus only on the assignment cost function $c(x, y_i) = -x_i$, which models our original resource allocation problem proposed in Section 3. Let $S = \{X^1, X^2, \ldots, X^m\}$ be $m$ independent samples from $\alpha$. The empirical version of the dual objective (11) is:

$$\mathcal{E}_S(g, \gamma) = \frac{1}{m}\sum_{t=1}^{m} \min_{i \in [n]} \bar{g}_{\gamma,c}(X^t, y_i) + g^{\top}\beta - \sum_{j,k,j \neq k} \gamma_{jk}\lambda_j \tag{16}$$

Let $\hat{g}_S, \hat{\gamma}_S$ be the empirical maximizer given the set of samples $S$: $(\hat{g}_S, \hat{\gamma}_S) := \arg\max \mathcal{E}_S(g, \gamma)$, and $g^*, \gamma^*$ be the population maximizer $(g^*, \gamma^*) = \arg\max \mathcal{E}(g, \gamma)$. We want to bound the number of samples needed so that $\mathcal{E}(g^*, \gamma^*) - \mathcal{E}(\hat{g}_S, \hat{\gamma}_S)$ is small with high probability. Let's introduce some notations to facilitate our later discussions. Define the following hypothesis class for each $i$:

$$F_i = \left\{ x \mapsto \bar{g}_{\gamma, c}(x, y_i) + g^\top \beta - \sum_{j,k,j\neq k} \gamma_{jk}\lambda_j : g \in \mathbb{R}^n, \gamma \in \mathbb{R}_+^{n(n-1)} \right\}. \tag{17}$$

as well as the overall hypothesis class:

$$F = \left\{ x \mapsto \min_{i\in[n]} \bar{g}_{\gamma, c}(x, y_i) + g^\top \beta - \sum_{j,k,j\neq k} \gamma_{jk}\lambda_j : g \in \mathbb{R}^n, \gamma \in \mathbb{R}_+^{n(n-1)} \right\}. \tag{18}$$

Plugging $c(x, y_i) = -x_i$ into the definition of $\bar{g}_{\gamma,c}$, we see that for a given $g$, and $\gamma$, the corresponding hypothesis $f_i \in F_i$ can be written as $f_i(x) = w^\top x + b$, where

$$w_j = \begin{cases} -(1 + \sum_{k\neq i} \gamma_{ik}), & \text{if } j = i \\ \gamma_{ji}\frac{\beta_j}{\beta_i}, & \text{if } j \neq i \end{cases}, \quad \text{and} \tag{19}$$

$$b = -g_i + g^\top \beta - \sum_{j,k,j\neq k} \gamma_{jk}\lambda_j. \tag{20}$$

It also follows that

$$F \subseteq F_{min} := \{x \mapsto \min_i f_i(x) : f_i \in F_i\}. \tag{21}$$

Note that $F$ defined in (18) is the main object of interest, as it contains all the possible Laguerre cell parameters. We showed in (21) that $F$ is at most as complex as $F_{min}$, a hypothsis class constructed from $n$ affine hypothesis classes. This interpretation of the original hypothesis class as the minimum over $n$ affine hypothesis classes is the key observation to prove the sample complexity bound. We prove our main result under the following boundedness assumption:

**Assumption 1.** *The hypothesis $f(x) = \min_i f_i(x) = \min_i {w^i}^\top x + b^i$ corresponding to the optimal dual solution $g^*, \gamma^*$ satisfies $\|w^i\|_1 \vee |b^i| \leq R$ for some $R > 0$. In particular, these assumptions imply that $F_i$ and $F$ are uniformly bounded by $R\bar{x} + R$.*

From (19) and (20) we can see that this is essentially a bound on the optimal dual variables $g^*, \gamma^*$, and a bound on the ratio $\beta_j/\beta_i$, both of which are determined by the input distributions $\alpha, \beta$, and do not depend on the number of samples. In other words, $R$ is a problem dependent constant.

**Theorem 2.** *Under Assumption 1, for a given sample size $m$, with probability $1 - \delta$, $\mathcal{E}(g^*, \gamma^*) - \mathcal{E}(\hat{g}_S, \hat{\gamma}_S) < O\left(\sqrt{\frac{(\log m)^3 + \log(1/\delta)}{m}}\right)$.*

The proof of this theorem can be found in the appendix. We will provide some high level intuition here. The proof uses the fat-shattering dimension, which is a concept that generalizes the Vapnik–Chervonenkis (VC) dimension to functions of real values. Like the VC dimension, the fat-shattering dimension is a measure of capacity for a class of functions. The larger the fat-shattering dimension, the more "complex" the function class is. Intuitively, the more complex the function class is, the more samples one needs to accurately identify the optimal function within that class. Recall that the object of interest in our paper is the Laguerre cell, whose boundaries are defined by a set of hyperplanes. Readers readers familiar with traditional PAC learning results might recall that hyperplanes have a low VC dimension. It turns out that the hypothesis class associated with Laguerre cells defined above in (18) has a low complexity as well. Similar to how low VC dimension leads to low sample complexity in classification tasks, the fact the boundaries of our Laguerre cells are consisted of hyperplanes also lead to low sample complexity in our setting.

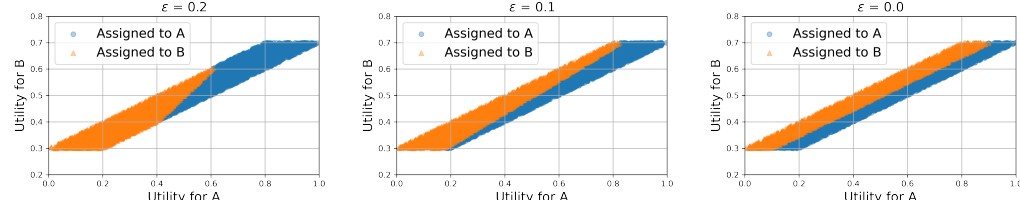

Figure 2: Allocation policy for artificial data under different envy constraints. From left to right: $\epsilon = 0.2, 0.1, 0.0$. When the envy constraint is loose (large $\epsilon$), $B$ envies $A$, since both agents prefer the items on the top right, but most of them are allocated to $A$. As the envy constraint tightens, the allocation boundary tilts in the direction that makes the allocations more even between the two agents.

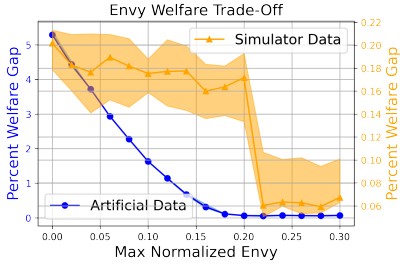

Figure 3: The trade-off curve between envy and welfare for both data-sets. The shaded region is between 25th and 75th percentile of the trials. The non-monotonicity in the plot for the simulator data is due to the stochasticity in the SGD algorithm.

Figure 4: Approximation gap with respect to sample size. Both $x$ and $y$ axis are in log scale. The solid line is the median and the shaded region is between the 25th and 75th percentile. The dashed lines show what the theoretical $1/\sqrt{m}$ rate would look like.

**Relation to Section 5** Note that there is some connection between the convergence of a stochastic optimization method (computational complexity), and the sample complexity bound of the same problem (learning complexity). Both say something about the number of steps/samples one needs to arrive at a good solution. However, in general, these two things are not the same. In particular, the learning complexity result from this section is algorithm agnostic. The proof of Theorem 2 actually implies uniform convergence: that no matter which hypothesis one considers, its' empirical objective will be close to the expected objective. However, Theorem 2 does not provide a way for practitioners to actually compute the solution. The result from Section 5 on the other hand does provide an algorithm for practitioners to use, and shows that the algorithm is computationally efficient. However, the convergence result in Section 5 only works for the specific optimization method that we proposed. Therefore these two results are complements of each other, and together paint a relatively complete picture on how difficult the task it.

## 7 Experiments

We test our solution with both artificial data, and simulated data from a realistic simulator for blood donor matching developed by [21]. The artificial data contains two recipients, and their valuation distribution is a linearly transformed uniform distribution. This is to make visualization of the resulting allocation policy easier. The simulator data is based on geographical and population information from San Francisco, and contains 5 recipients. To set the envy budgets $\lambda \in \mathbb{R}^n$, we first decide on a constant $\epsilon \in \mathbb{R}_+$, and then multiply this by the target matching distribution $p^* \in \mathbb{R}^n$: $\lambda_{ij} = \epsilon p_i^* \forall i, j$. With this setup $\epsilon$ is a bound on the normalized envy for each recipient: $\frac{1}{p_i^*} Envy(i) \leq \epsilon \forall i$. Figure 2 illustrates how the allocation policy changes as we change $\epsilon$. As the envy constraint tightens, the decision boundary tilts in the direction that split the "good" (items which both agents prefer) and "bad" (items which both agents dislike) items more evenly between the recipients.

Next we investigate the trade-off between envy and social welfare by using SGD to compute approximately optimal allocations for varying $\epsilon$. We plot the percent welfare gap (difference between the

maximum welfare without envy constraints, and the welfare with envy constraints, divided by the former) with respect to realized, max normalized envy. Figure 3 shows the result. For the simulator data, the welfare gap is small even with a no-envy constraint, which means that aiming for envy free allocations might make sense. In the case of the artificial data however, paying $50\%$ of the full price of fairness reduces $65\%$ of the envy. In such settings, one might want to sacrifice some envy for better welfare.

These experiments also highlight when envy arises. When recipients' utilities are highly correlated, but one recipient has larger variance than others, that recipient receives almost all the good items (which results in large envy for other recipients), even though others value the items almost as much. In such cases, a small reduction in welfare can reduce a large amount of envy. This seems to be the case for the simulator data. On the other hand, if utilities are correlated, but only one recipient has very strong preferences, then allowing a small amount of envy can improve the welfare significantly.

Finally, in Figure 4 we investigate the quality of the empirical solutions as the sample size increases. It can been seen that the approximation gap decreases faster than the theoretical rate, confirming our sample complexity bound in Theorem 2.

## 8   Limitations and Future Directions

Although we believe that the model proposed here is natural, and captures the most salient aspects of some of the resource allocation problems in real life, any implementation of our proposed strategy in critical applications such as blood donation should be prefaced with more rigorous backtesting in order to minimize the risk of unintended consequences in application specific metrics not studied in this paper.

For future directions, one key property that we did not study in this paper is the problem of incentive compatibility. For our motivating application of blood donation, this is not an issue because online platforms such as Meta has proprietary models that can predict the matching quality between donor and recipient. This means that the platform observes the value of matchings without having to rely on the recipients to self-report. This is also true in many other online matching problems such as sponsored ads. However, in settings where the central planner relies on the recipients to self-report their valuations for each of the items, incentive compatibility becomes a crucial issue. We are excited about the potential of using Optimal Transport in fair-division, and plan on exploring the incentive issues in future work.

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
