# A   Auxiliary Proofs

## A.1   Proof of Theorem 1

*Proof.* Using Fenchel-Rockafellar's duality theorem, the dual of (7) can be written as

$$\max_{f,g,\gamma \geq 0} \int_{\mathcal{X}} f(x) d\alpha(x) + g^{\top}\beta - \sum_{j,k,j \neq k} \gamma_{jk}\lambda_j \tag{22}$$

$$s.t. \quad \left(1 + \sum_{k \neq j} \gamma_{jk}\right) c(x,j) - \sum_{k \neq j} \gamma_{ky} c(x,k) \frac{\beta_k}{\beta_j}$$

$$- f(x) - g_j \geq 0 \quad \forall x \in \mathcal{X}, y_j \in \mathcal{Y}$$

Fixing $g \in \mathbb{R}^n$ and $\gamma \in \mathbb{R}^{n(n-1)}$, we can check using first order conditions that the optimal $f(x)$ has the closed form expression:

$$\min_{j \in [n]} \bar{g}_{\gamma,c}(x,y_j) := \left(1 + \sum_{k \neq j} \gamma_{jk}\right) c(x,j) - \sum_{k \neq j} \gamma_{kj} c(x,k) \frac{\beta_k}{\beta_j} - g_j$$

Using this, the infinite dimensional optimization problem in (22) can be transformed to a finite dimensional optimization problem:

$$\max_{g,\gamma \geq 0} \mathcal{E}(g,\gamma) := \int_{\mathcal{X}} \min_{j \in [n]} \bar{g}_{\gamma,c}(x,y_j) \, d\alpha(x) + g^{\top}\beta - \sum_{j,k,j \neq k} \gamma_{jk}\lambda_j \tag{23}$$

Alternatively, we can adapt the Laguerre cell notation in (2) to (23):

$$\mathcal{E}(g,\gamma) = \sum_{i \in [n]} \int_{\mathbb{L}_{y_i}(g,\gamma)} \bar{g}_{\gamma,c}(x,y_i) d\alpha(x) + g^{\top}\beta - \sum_{j,k,j \neq k} \gamma_{jk}\lambda_j$$

where $\mathbb{L}_{y_i}(g,\gamma) = \left\{ x \in \mathcal{X} : y_i = \arg\min_{y_j} \bar{g}_{\gamma,c}(x,y_j) \right\}$. □

## A.2   Proof of Theorem 2

*Proof.* We prove the result via uniform convergence:

$$\mathcal{E}(g^*,\gamma^*) - \mathcal{E}(\hat{g}_S, \hat{\gamma}_S)$$
$$= \mathcal{E}(g^*,\gamma^*) - \mathcal{E}_S(\hat{g}_S, \hat{\gamma}_S) + \mathcal{E}_S(\hat{g}_S, \hat{\gamma}_S) - \mathcal{E}(\hat{g}_S, \hat{\gamma}_S)$$
$$\leq \mathcal{E}(g^*,\gamma^*) - \mathcal{E}_S(g^*,\gamma^*) + \mathcal{E}_S(\hat{g}_S, \hat{\gamma}_S) - \mathcal{E}(\hat{g}_S, \hat{\gamma}_S)$$
$$\leq \sup_{g,\gamma} \left( \mathcal{E}(g,\gamma) - \mathcal{E}_S(g,\gamma) \right) + \sup_{g,\gamma} \left( \mathcal{E}_S(g,\gamma) - \mathcal{E}(g,\gamma) \right)$$
$$\leq 2 \sup_{g,\gamma} |\mathcal{E}(g,\gamma) - \mathcal{E}_S(g,\gamma)| \tag{24}$$

Clearly, it suffices to show that $\mathcal{E}_S(\cdot)$ converges uniformly to $\mathcal{E}(\cdot)$. For a given $g,\gamma$, the dual objective function and its' empirical version can be written as

$$\mathcal{E}(g,\gamma) = \mathbb{E}_{\alpha}[f(X)], \quad \mathcal{E}_S(g,\gamma) = \frac{1}{m}\sum_{t=1}^{m} f(X^t).$$

Then we can rewrite the supremum in (24) as:

$$\sup_{g,\gamma} |\mathcal{E}(g,\gamma) - \mathcal{E}_S(g,\gamma)| = \sup_{f \in F} \left| \mathbb{E}_{\alpha}[f(X)] - \frac{1}{m}\sum_{X \in S} f(X) \right| \tag{25}$$

Since $|f(X)| \leq (R\bar{x} + R)$ for all $f \in F, X \in \mathcal{X}$, it follows from Theorem 26.5 in [26] that with probability $1 - \delta$,

$$\sup_{f \in F} \mathbb{E}_\alpha[f(X)] - \frac{1}{m} \sum_{X \in S} f(X) \leq 2\mathbb{E}_S\left[\text{Rad}_m(F \circ S)\right] + (R\bar{x} + R)\sqrt{\frac{2\log(2/\delta)}{m}} \quad (26)$$

and the same also holds by replacing $F$ with $-F$. Here

$$\text{Rad}_m(F \circ S) := \mathbb{E}_\sigma\left[\frac{1}{m} \sup_f \sum_{j=1}^m \sigma_j f(X_j)\right]$$

is the standard definition of Rademacher complexity of the set $F \circ S$. Since $\sigma_i$ are $i.i.d.$ Rademacher random variables, it is easy to see that $\text{Rad}_m(F \circ S) = \text{Rad}_m(-F \circ S)$. Therefore we can use a union bound to obtain that with probability $1 - \delta$,

$$\sup_{f \in F} \left|\mathbb{E}_\alpha[f(X)] - \frac{1}{m} \sum_{X \in S} f(X)\right| \leq 2\mathbb{E}_S\left[\text{Rad}_m(F \circ S)\right] + (R\bar{x} + R)\sqrt{\frac{2\log(4/\delta)}{m}} \quad (27)$$

It remains to bound the Rademacher complexity of the $F \circ S$. To do so, we use tools from learning theory, and give the following bound on the fat-shattering dimension ([8]) of the hypothesis class $F$.

**Lemma 1.** *Under Assumption 1, $F$ has $\zeta$-fat-shattering dimension of at most $\frac{c_0(R\bar{x}+R)^2}{\zeta^2} n\log(n)$, where $c_0$ is some universal constant.*

The proof of Lemma 1 can be found in the Appendix. The above bound on the fat-shattering dimension can be used to bound the covering number (see Definition 27.1 of [26]) of $F \circ S$. Theorem 1 from [22] states that

$$\mathcal{N}(\delta, F, ||\cdot||_2) \leq \left(\frac{2B}{\delta}\right)^{c_1 \text{fat}_{c_2\delta}(F)} \quad (28)$$

where $B$ is a uniform bound on the absolute value of any $f \in F$. Let $B = (R\bar{x} + R)$, we have that

$$\text{Rad}_m(F \circ S)$$
$$\leq \inf_{\delta'>0}\left\{4\delta' + 12\int_{\delta'}^B \sqrt{\frac{\log\mathcal{N}(\delta, F, ||\cdot||_2)}{m}}d\delta\right\}$$
$$\leq \inf_{\delta'>0}\left\{4\delta' + 12\frac{\sqrt{c_1 c_0}}{c_2}B\sqrt{\frac{n\log n}{m}}\int_{\delta'}^B \sqrt{\log\left(\frac{2B}{\delta}\right)}d\delta\right\}$$
$$= c'\sqrt{\frac{n\log n(\log m)^3}{m}}$$

Where we used Dudley's chaining integral [28, 16], Lemma 1 and (28), and setting $\delta' = \frac{1}{\sqrt{m}}$ respectively. Plugging the above back to (27) and (24), we see that with probability $1 - \delta$,

$$\mathcal{E}(g^*, \gamma^*) - \mathcal{E}(\hat{g}_S, \hat{\gamma}_S) \leq c'\left(\sqrt{\frac{n\log n(\log m)^3}{m}} + \sqrt{\frac{1\log\frac{1}{\delta}}{m}}\right).$$

Conversely, ignoring the log terms, $m$ needs to be at most on the order of $\tilde{O}\left(\frac{n}{\epsilon^2}\right)$ in order for $\mathcal{E}(g^*, \gamma^*) - \mathcal{E}(\hat{g}_S, \hat{\gamma}_S)$ to be bounded by $\epsilon$ with high probability.

$\square$

Proof of Lemma 1

*Proof.* Theorem 3 in [20] shows that $\text{fat}_\zeta(F_{min}) \leq \frac{c_0(R\bar{x}+R)^2}{\zeta^2} n\log n$. Since the shattering dimension is monotone in the size of the set, we are done.

$\square$

## A.3 Experimental Setup

For the artificial data, the value utility vectors are generated from $X = [1, 0.7] - Z \begin{bmatrix} 0.2, & 0 \\ 0.8, & 0.4 \end{bmatrix}$ where $Z \sim Unif(0, 1) \times Unif(0, 1)$. For finding the optimal allocation policy on the artificial data, we used Algorithm 1 with $T = 2 \cdot 10^5$. For simulator data, we used $T = 2.5 \cdot 10^6$. To generate Figure 2 we sampled 6000 points from the distribution and plotted them, colored by the allocation. For Figure 4, for each $m$ we ran 16 trials, sampling a different set of $m$ data points as our training data per trial. All experiments are run on a 2019, 6-core Macbook Pro laptop. The simulator code is open sourced by [21] at https://github.com/duncanmcelfresh/blood-matching-simulations, and also included in the supplementary material.