# OpenReview forum: "Optimal Efficiency-Envy Trade-Off via Optimal Transport"
_NeurIPS.cc/2022/Conference — NeurIPS 2022 Accept_

### Official Review · Reviewer_yMJc · 2022-07-03

**Rating:** 5
**Confidence:** 3
**Soundness:** 4 excellent
**Presentation:** 2 fair
**Contribution:** 2 fair

**Summary:**

Here the authors consider the problem of allocating a distribution of items to $n$ fixed individuals (named "receivers") subject to a fixed fraction constraint, while simultaneously ensuring minimal envy.  At a high level, the authors show that this problem can be formulated as a variant of the optimal transport problem subject to a particular cost function (and the fact that "one" if the distributions in the problem is discrete).  The authors then show convergence guarantees for an empirical version of stochastic gradient descent using observed samples from the underlying distribution, and complement their results with a numerical simulation on a synthetic and practically motivated dataset.

To be more concrete, the authors consider a fixed set $n$ of individuals $Y$.  There is an "infinite" number of items represented by a distribution $\alpha$ over $[0,1]^n$ where each draw from this distribution $\alpha$ is a vector representing each individual $i$'s utility for the particular item.  The goal is to maximize the expected utilities of the individuals, while maintaining the constraint that each individual $y_i$ is matched at least $p_i$ of the items in expectation.

The authors consider allocation policies $\pi$ which takes in a valuation vector (i.e. the $n$ dimensional vector of individual utilities for the item) and maps it to one of the $n$ receivers.  The basic formulation is then to solve the optimization problem to maximize the $\sum_i X_i \pi(y_i | X)$ subject to the constraint that $\Pr(\pi(y_i | X)) \geq p_i$.  Note that the objective function maximizes the utilitarian welfare, and the constraint ensures the target matching constraint.  However, one (potentially) additional desired property is envy-freeness.  This can be incorporated by allowing a tolerance $\lambda_i$ of envy for each individual, and adding the constraint that $\max_j u_i(\pi_j) - u_i(\pi_i) \leq \lambda_i$ where we drop some notation and use $u_i(\pi_j)$ to denote the expected utility for $i$ given $j$'s allocation.

The main result of the authors is three-fold under this model:
1. Connection to Optimal Transport: The authors show how while the feasible solution space to this problem is large, tools from OT and Fenchel Duality can be used to show a relationship between Optimal Transport (under a negative linear cost function) to an optimal solution.  In particular, the authors show that the optimal solution is given as a greedy function over a partition (called "Laguerre cells").
2. Practical Optimization: The authors then show that the dual of the optimization problem (in the OT formulation) can be solved either with stochastic gradient descent or samples.  With this the authors show a high probability convergence guarantee on the utilitarian welfare scaling as $1 / \sqrt{m}$ where $m$ is the number of samples.
3. Experiments: The authors complement the theoretical results with an experiment on a synthetic dataset and a practically motivated one, highlighting the empirical loss in efficiency as one modifies the fairness constraints and empirical convergence rates.

**Questions:**

### Questions
1. The comparison between section 5 and section 6 potentially could use more emphasis.  In Section 5 the authors highlight how stochastic gradient descent with samples (unbiased from the true underlying distribution) achieve a particular convergence guarantee.  However, in section 6, this is then done again via a slightly different method, is the only intuitive difference being obtaining high probability versus in expectation guarantees?
2. Frequently in the paper the authors mention that their work addresses the welfare cost for achieving envy-free solutions.  Are there any theoretical results supporting this claim, highlighting price of fairness under this model?

### Minor Comments
- In paragraph , "Unlike the existing resource allocation literature, where envy-free is treated as a hard constraint or not considered at all" is not totally true with the ongoing literature studying EF1/EPX, etc with divisible resources, (Donahue + Kleinberg 2019), etc.
- The second bullet in line 51 could be expanded more
- The "proof by picture" in Figure 1 should be expanded more


**Limitations:**

In Section 8 the authors properly address the limitations and potential negative social impact of their work - notably being that any implementation of the strategy for designing allocation mechanisms must be used carefully in critical scenarios such as blood donation settings.

**Strengths And Weaknesses:**

### Strengths
1. Model + Theoretical Results: The theoretical results presented in section 4 is novel to the fair resource allocation literature, by making an explicit connection between allocation policies and optimal transport literature.  As the authors state, this avenue leads to interesting future directions both theoretically and practically as more tools from OT are used in the literature.  In particular, to some extent, the model presented can be thought of as an "infinite resource" version of many exiting literature, which typically assumes that the distribution governing utility of individual is also discrete.
2. Quality of Results: The theoretical results presented in section 5 + 6 help highlight the ability to use the dual of the OT formulation practically to an algorithm design.
3. Relation to Existing Literature: The authors do a good job relating the current analysis to the optimal transport literature, especially for readers in the fairness community who would be less aware of this line of work.

### Weaknesses
1. Weakness of Results: One aspect that the authors mention frequently in their paper is their ability to understand the "optimal tradeoff" between envy and efficiency.  While the OT formulation allows them to optimally solve for the solution given the additional envy constraints, there is no theoretical results or justification on the "efficiency" loss one observes when adding the additional envy constraints.  There are numerical simulations highlighting this aspect, but the theoretical results are lacking in that regard.  Outside of formalizing the relationship between allocation policies and optimal transport, the authors provide straightforward ML generalization guarantees and SGD formulation.
2. Relation to Existing Literature: The authors need to improve on situating their paper in the fair resource allocation literature.  In particular, some model primitives (i.e. ex-ante vs ex-post allocations, relation to the "price of fairness" eg (Donahue + Kleinberg, 2019), etc need to be better made clear after describing the model (more of this in next bullet).
3. Writing + Practical Motivation: The authors do an amazing job highlighting the connection between the optimal transport literature and fair resource allocation.  However, the practical description of their model and relation to previous literature on fair resource allocation needs to be expanded more.
- Model Description: The paper could use a more concrete model description, in particular substantiating some of the model parameters in the running blood matching example.  For example, in line 40/41 are the receivers here the blood bank vs. individual users of the blood blank matching platform?
- "Target Matching Distribution": The target matching distribution $p_i^\star$ is introduced with no practical motivation.  It leaves questions open like why the matching policy only needs to satisfy these constraints in expectation instead of almost surely (where then the problem is trivial), how they should be chosen, and their relationship to the envy-free guarantee which is described later.
- Ex-Ante vs Ex-Post Allocations and the Online Structure: The matching policy solved for $\pi$ denotes the probability of matching an item to $y$ given a specified valuation vector $X$ (denoting the utility of the $n$ individuals for the given item).  Consider an application of this policy: an item is observed (hence an $X$), and the item is matched or shared at the discrete fractions $\pi(y | X)$.  However, notably this does not satisfy the target matching distribution (although...if those constraints were added almost surely the problem would be trivial).  However, the hindsight solution (i.e. once $X$ is observed) is trivial, either allocate based on the fractions $p_i^\star$, or allocate to the individual who cares the most, or subject to some envy-freeness constraints guaranteed by $\lambda$.  The motivation for considering these "ex-ante" allocations is never explicitly given.  Moreover, the objective problem is only situated to consider a "single" item drawn from the infinite set of items, instead of considering potentially multiple allocations (although this is somewhat abated by the expectation).  The relationship between the model considered here and the related literature (e.g. Donahue + Kleinberg highlighted earlier and the literature therein), or a discussion on the relationship between the blood bank problem should be included.

---

> ### Author Response · Authors · 2022-08-02
> **Thank you for your comments.**
>
> Thank you for your comments. We will respond to the weaknesses and the questions you raised in order.
>
> Weakness 1 (weakness of results): This is a fair point. Although our formulation allows practitioners to optimally trade-off between efficiency and envy, our results do not contain theoretical predictions on how “severe” the trade-off has to be under different distributions. It would be interesting to characterize exactly which types of distributions lead to more envy, and what the shape of that trade-off curve might be. We do think however, that the results provided are already interesting by themselves, especially for practitioners.
>
> Weakness 2 (relation to existing literature): Thank you for bringing this to our attention. Donahue & Kleinberg 2020 does indeed also study a setting where the fairness criteria can be smoothly adjusted with respect to welfare. However their setting is quite different from ours, since they assume that there is a single type of resource where the only thing the agents care are the amount/probability of receiving the items. In our revision, we have added a clearer discussion of ex-ante vs ex-post fairness, how it relates to Donahue & Kleinberg (and others), and why we focus on ex-ante fairness. See also our response below.
>
> Weakness 3 (Writing + practical motivation): Regarding relation to previous literature on fair division, we have added additional discussion on that in the literature review section.
>
> Weakness 4 (Model description): The receivers here are the blood banks. The “items” being allocated in this case are the potential blood donors. We will revise the exposition to make this clearer.
>
> Weakness 5 (Target matching distribution): If we did not have a target matching distribution, and the objective is simply to maximize social welfare, then we might end up with an allocation where some donation centers receive 0 donors (for instance, those that are located in more rural areas) and others receive all the donors. Intuitively, this is not a result that we want (rural donation centers might serve a lot of patients). Introducing the target distribution allows the central planners to fix the ratios so that each blood-bank receives some fraction of the donors. This is useful (separately from envy freeness)  if the platform wants to convince recipients to participate on the platform, by guaranteeing that they will be recommended a certain fraction of the time.
>
> Weakness 6 (Ex-Ante vs Ex-Post allocation): The reason why we consider ex-ante guarantees has to do with the type of application we're interested in. Imagine that you are performing blood donations recommendation on an internet platform. In such settings, we are assigning tens, and even hundreds of millions of users to blood banks. Even if we want them to hold ex-post, the large-scale nature of the problem + the law of large numbers means that in-expectation guarantees translate into something that is very close to holding ex-post. That is why for internet platform problems, requiring constraints to hold in expectation is a pretty standard setup. See for example the literature on budget constraints in ad auctions, where this is the case (see for example Balseiro, Besbes, Weintraub 2014) The reviewer might have mistakenly understood our setup as allocating a single item. We are in fact interested in allocating all items as represented by the distribution.
>
> Question 1: The reviewer is right to point out that there is some connection between the convergence of a stochastic optimization method (computational complexity), and the sample complexity bound of the same problem (learning complexity). In general, these two things are not the same. In particular, the learning complexity result from Section 6 is algorithm agnostic. In fact, it also shows uniform convergence, meaning that no matter which solution one arrives at, the empirical objective is close to the expected objective. However, the result from section 6 does not provide a way for practitioners to actually compute the solution. The result from section 5 on the other hand does provide an algorithm for the practitioner to use, even though the convergence result only works for the specific optimization method that we proposed.
>
> Question 2: Given the envy constraint specified by the central planner, the allocation proposed by our formulation is optimal – meaning that there is no other allocation with at most the same amount of envy, satisfies the target distribution constraints, and achieves higher social welfare. This is simply guaranteed by the constrained optimization formulation. Caragiannis et al. 2009 provide some worst case analysis on how much welfare one has to lose to achieve envy-freeness. Our formulation allows central planners to smoothly trade-off envy and efficiency, instead of having to choose between either zero-envy or maximum envy.
>
> Minor Comments: We added a discussion of EF1/EFX in the literature review section as well.

---

> > ### Comment · Reviewer_yMJc · 2022-08-06
> > **Response to comments**
> >
> > Thanks for the detailed response to all of the comments!  Unfortunately my main point of critique is with the exposition, where any modifications are difficult to verify under the NeurIPS review format.
> >
> > The writing is very technical, which for a paper who's (to my understanding) main thesis is using techniques from the optimal transport field and applying it to the fair division literature, could do a better job explaining the fair division model and contrasting it to the current literature to highlight the main ideas.
> >
> > Weakness 1: Agreed - I think potentially the tone of some of the writing around these facets could be reduced more to highlight it from a "practical" point of view on allowing practitioners to adjust these parameters, without providing a full characterization on the trade-off.
> >
> > Weakness 6: Right, exactly.  I think that some of the exposition around this fact should be included in the paper.  Especially when one considers what the "policy" is - a mapping from a single item to the set of recipients.  Deriving ex-post fairness guarantees under this setting probably doesn't make much sense, which is why the ex-ante where the "policy" is evaluated several times is more intuitive.

---

> > > ### Author Response · Authors · 2022-08-06
> > > **Thank you for your comments**
> > >
> > > Thank you for your comments. In our revision (already uploaded to the OpenReivew submission website), we moved the more technical proofs in Section 6 to the appendix, and used the additional space for a discussion on the choice of ex-ante constraint, more discussions in the literature review section, as well as a few other things that other reviewers brought up.

---

### Official Review · Reviewer_ndyP · 2022-07-09

**Rating:** 5
**Confidence:** 3
**Soundness:** 4 excellent
**Presentation:** 2 fair
**Contribution:** 3 good

**Summary:**

The paper studies a fair division problem inspired by Facebook’s blood donation program: a continuum of divisible goods must be allocated to “receivers” with additive utilities in such a way that (1) each receiver $i$ receives a target fraction $p_i$ of all goods, such that (2) each receiver $i$ has envy below some given bound $\lambda_i$, and such that, within these constraints, (3) utilitarian welfare is maximized. The paper reduces this problem to a form of optimal transport, from which they derive results about the structure of optimal allocations, a convex optimization formulation, a stochastic gradient descent algorithm based if the distribution of goods can only be sampled rather than being explicitly given, and a PAC-style learning algorithm if a finite number of samples are available. The authors test these algorithms on synthetic and semi-synthetic data, and empirically investigate the price of tighter envy bounds in terms of welfare.

**Questions:**

No specific questions, but please feel free to respond to anything you’d like.

**Limitations:**

No complaints.

**Strengths And Weaknesses:**

I was pleasantly surprised by the results of this paper. I wasn’t familiar with optimal transport, and would have assumed a priori that the optimization problem solved in the paper would lead to quite messy solutions and approaches. But the authors show that by using the right kind of duality and (what sounds in their writing like standard) approaches from optimal transport and learning theory can make quick inroads into this problem. Given that I am not an expert, I cannot judge the novelty of the technical contribution, but they were new to me.
I also really appreciate the empirical evaluation, which studies three well-selected questions in little space. Each of the three figures on page 8 tells an interesting story.

My main point of critique is the exposition, which is very math-heavy and not particularly accessible to an outside audience. In some sense, I think that this is the drawback of my first positive point: some of the paper’s value comes from bringing techniques from other fields into fair division, but the paper could do a better job at actually speaking to a fair-division audience where these tools aren’t yet common knowledge. I want to exclude Section 2.1 from this critique, which did a decent job at laying out the basic background of Optimal Transport. By contrast, a lot of the technical material references so many results from existing papers with so little description that I couldn’t take much intuition from it. I also think that the analysis of Algorithm 1 (stochastic gradient descent) should be proved in the supplementary material. More generally, part of the paper consists of more formulas than text, which makes following along much more painful than it would have needed to be. I would strongly recommend putting more of the technical details into the appendix and trying to convey in words more of an intuition of how the argument proceeds and why it works.

## Minor comments:

(1) To me, the word “receiver” sounds like radio terminology, not fair division. In case this is not already established language, how about “recipient”? At a quick glance, McElfresh et al. seem to also use this word.

(2) Equations like Eq. 2 would be much easier to parse with parentheses.

(3) I was going to complain about the missing derivation of the Fenchel dual, but saw it in Appendix A.1, which isn’t mentioned anywhere in the draft. I’d encourage the authors to change that!

---

> ### Author Response · Authors · 2022-08-02
> **Thank you for your comments.**
>
> Thank you for your comments. In our revision, we have expanded our discussion on the existing fair division literature to help readers better position our paper against other work. For sections 4 (solution structure) and 7 (experiments), we focused a lot of the discussions on building intuitions on how to interpret the results.  For sections 5 and 6 however, the results are more technical, which means that there are more equations. We have added more discussions in the beginning of these sections to make the results more interpretable.

---

> > ### Comment · Reviewer_ndyP · 2022-08-03
> > **Rebuttal response**
> >
> > Thank you for your response. Based on what I have seen, your revision doesn’t yet address what bothered me about the exposition. (What you added to the introductions of Section 5 & 6 is fine by me, but nothing I was missing as a reader.) Let me try again, in a more pointed way:
> >
> > For me as a reader, it seems like your not using space effectively in Section 6. Let me demonstrate this with the second half of the proof of Thm. 2, after Lemma 1. At this point, you are using the fat-shattering dimension (not defined in the paper, only in the reference), the covering number (not defined, only in reference), Thm. 1 from [23] (not restated), and Dudley’s chaining integral (not defined, only in reference). Even though I know traditional PAC learning, I derived no insights from this half page of text, and this is just one example. I can’t tell whether there is some specialist audience for which this proof will be super valuable to have in the body, but I think it’s a very valid question whether you would not get more mileage out of the paper by deferring the proof for specialists in the appendix (and perhaps making it less terse there) and filling this space with anything else that you feel is important to get across.

---

> > > ### Author Response · Authors · 2022-08-06
> > > **Thank you for your suggestion**
> > >
> > > Thank you for the suggestion. We agree that the proof details in Section 6 can be relegated the appendix. We have moved it to the appendix, and used the additional space for a) a high level summary of the proof in section 6, b) additional discussions of the relationship between the results in Section 5 and 6, which hopefully will help readers with understanding the results better. c) additional discussions around the modeling assumptions which some other reviewers had confusions about, and d) more discussions in the literature review section.

---

### Official Review · Reviewer_3mgQ · 2022-07-10

**Rating:** 7
**Confidence:** 3
**Soundness:** 3 good
**Presentation:** 2 fair
**Contribution:** 3 good

**Summary:**

The key contribution of this paper is the formulation of the resource allocation problem with envy restrictions as a variant of the semi-discrete optimal transport problem, which can thus be solved using the projected SGD algorithm.  The paper also shows some statistical properties that may be helpful for its real world applications.


**Questions:**

Though the paper looks reasonable, there are still places that can be improved.
For example, the paper did not show clearly the relationship between conditional probability and joint probability in the optimal solution structure, which makes the problem formulation less strict. It would be more convincing if the paper can put these details into the appendix.


**Limitations:**

The paper solves the problem using some SGD method. Thus, from an algorithmic point of view, it does not provide any new method or improvement. Although the paper presents a new way to look at the envy-tolerance resource problem, it does not show  how the optimal transport properties play roles in the solution. Another limitation is that the appendix is not well written. Some deductions in the appendix
need to be clearer and more detailed.


**Strengths And Weaknesses:**

The strength (also novelty) of this paper is that it relaxes the resource allocation problem with envy-free restrictions to the one with envy-tolerance. It also proposes a semi-discrete optimal transport variant formulation.

Overall, this is a good paper and well organized. The assumptions and problems are explicitly stated. The novel formulation may provide the possibility of applying various optimal transport methods to envy-related resource allocation problems.
The SGD method is not new, but serves the purpose adequately.

---

> ### Author Response · Authors · 2022-08-02
> **Thank you for your comments.**
>
> Thank you for your comments. We agree that the results, when viewed from an algorithmic perspective, are not groundbreaking. However, we view our main contribution as providing a new framework for studying one of the most studied fairness constraints – envy-freeness, by relaxing it from a hard constraint,  to one that can be smoothly adjusted, along with the connection to optimal transport. In addition, we provide the analysis of a practical optimization method, as well as an analysis of the statistical properties of the solution structure, thus providing a comprehensive treatment of the problem. We believe that this is a meaningful contribution to the fair division community, and hope that it will catalyze more research effort in the intersection of fair division and optimal transport.

---

> > ### Comment · Reviewer_3mgQ · 2022-08-07
> > **Thank you for your comments**
> >
> > Thank the authors for the comments, which confirm my original thoughts about this paper. For this reason, I will stick to my positive rating of this paper.

---

### Official Review · Reviewer_GLsA · 2022-07-11

**Rating:** 7
**Confidence:** 2
**Soundness:** 4 excellent
**Presentation:** 3 good
**Contribution:** 3 good

**Summary:**

This paper considers the resources allocation problem where each of $n$ receivers has to be allocated a fixed fraction of items. For each round an item is drawn from a distribution $\mathcal{D}$. Each item can be represented by a vector $x$ in $\mathbb{R}^n$, where $x_i$ is $i$-th receiver's value of this item. In this model the optimal allocation rule that maximizes social welfare can be model as the solution to a semi-discrete optimal transport (OT) problem. The solution structure and efficient algorithm via stochastic optimization are well-studied in the literature of OT.

The main contribution of this paper is that they study the tradeoff between envy and efficiency in this model. They add one envy constraint for each receiver: envy of receiver $i$ against any other receiver should be bounded by $\lambda_i$. The goal is to find a socially optimal allocation rule under these constraints. They apply Fenchel-Rockafellar duality to characterize the optimal solution, which is in a similar form to the original OT optimization problem without envy constraints. The paper shows that with slight modification, the stochastic optimization for the original OT problem is efficient for the new problem. Finally the authors provide a PAC-like sample complexity bound using standard Rademacher complexity argument.

The authors also provide some numerical simulation results of their framework.

**Questions:**

- Does not provide many real world applications of their model. The only application discussed in the paper is the blood bank donation. It would be great if the authors can provide more potential real world applications.


**Limitations:**

The authors adequately addressed the limitations and potential negative societal impact of their work.

**Strengths And Weaknesses:**

Strengths:
- The paper considers an important model of resource allocation. The tradeoff between envy and efficiency is also a natural question to ask in the setting.

- The paper studies this model comprehensively, it not only characterize the optimal tradeoff between envy and efficiency but also give a PAC-like sample complexity bound.

- The paper includes some numerical simulations.

Weaknesses:

- The literature review is a bit problematic. As far as I am concerned, some previous works have been discussed applications of optimal transport in resources allocation and mechanism design problems (see e.g. A. Galichon, Optimal Transport Methods in Economics). The authors should have discussed these results.

- As of technical parts, techniques and proofs are not very interesting in general. Everything seems to be standard.

---

> ### Author Response · Authors · 2022-08-02
> **Real world applications**
>
> Thank you for your comments. Regarding other real world applications of our formulation: our formulation applies to any setting where the recipients have different valuations for the different items, and where we want to ensure that each recipient receives a fraction of the items. Other potential examples include: allocating unsold groceries to food banks, and allocating ad impressions to advertisers in settings with pre-specified contracts on how many ad impressions each advertiser must receive.

---

> > ### Comment · Reviewer_GLsA · 2022-08-10
> > **Thanks for the response**
> >
> > Thank the authors for the response. I will keep my score unchanged. I encourage authors to do a more extensive literature review and compare the results and methods in the future version.

---

### Meta-Review · Area_Chair_rWM9 · 2022-08-26

**Recommendation:** Accept
**Confidence:** Less certain

**Metareview:**

Executive summary:

The problem considered in this paper is as follows: There is a distribution over items X \subseteq [0,\bar{x}]^n where x_i denotes the value of the item to recipient i. There are also matching constraints {p_i}_{i \in N}, which require that each agent should be matched a p_i fraction pf the times. The goals is to maximize the sum of recipient utilities subject to the matching propability constraints, and also satisfying that no recipient i envies another recipient by more than a factor \gamma_i.

It is shown that this problem can be solved as a semi-discrete optimal transport problem. They also give a stochastic optimization algorithm which converges at rate O(1/sqrt(T)), and a PAC-style sample complexity result (showing that with O(n/eps^2) samples an eps-approximate solution can be found with high probability).

Discussion and recommendation:

This paper is a bit out of my comfort zone, so I am mostly relying on the reviews, which are rather positive and supportive of the paper. The connection to optimal transport is appreciated, and the approximation results (while rather standard) seem to find their audience as well.

Weak accept.

**Award:**

No

---

### Decision · Program_Chairs · 2022-09-14

Accept